# Latrophilin-2 and latrophilin-3 are redundantly essential for parallel-fiber synapse function in cerebellum

Roger Shen Zhang[1], Kif Liakath-Ali[1], Thomas C Südhof[1,2]*

[1]Department of Molecular and Cellular Physiology, Stanford University, Stanford, United States; [2]Howard Hughes Medical Institute, Stanford University, Stanford, United States

**Abstract** Latrophilin-2 (Lphn2) and latrophilin-3 (Lphn3) are adhesion GPCRs that serve as postsynaptic recognition molecules in CA1 pyramidal neurons of the hippocampus, where they are localized to distinct dendritic domains and are essential for different sets of excitatory synapses. Here, we studied Lphn2 and Lphn3 in the cerebellum. We show that latrophilins are abundantly and differentially expressed in the cerebellar cortex. Using conditional KO mice, we demonstrate that the Lphn2/3 double-deletion but not the deletion of Lphn2 or Lphn3 alone suppresses parallel-fiber synapses and reduces parallel-fiber synaptic transmission by ~50% without altering release probability. Climbing-fiber synapses, conversely, were unaffected. Even though ~50% of total cerebellar Lphn3 protein is expressed in Bergmann glia, Lphn3 deletion from Bergmann glia did not detectably impair excitatory or inhibitory synaptic transmission. Our studies demonstrate that Lphn2 and Lphn3 are selectively but redundantly required in Purkinje cells for parallel-fiber synapses.

## Introduction

Brain function is thought to be mediated by information processing in neural circuits that in turn are constructed via formation of synaptic connections between neurons. Neural circuit wiring requires directed growth of axons towards target regions, self-avoidance of axons and dendrites, axon-dendrite target selection, and formation and specification of synapses (*Hassan and Hiesinger, 2015*; *Yogev and Shen, 2014*; *Zipursky and Sanes, 2010*). Although much is known about axon guidance, the molecular mechanisms governing target selection and synapse formation remain largely unclear.

Synaptic adhesion molecules form trans-synaptic complexes across the synaptic cleft that are thought to initiate synapse formation, maintain synapse stability, and regulate synapse properties (*Südhof, 2018*). Many synaptic adhesion molecules have been identified, of which latrophilins and BAIs may be unique because these adhesion GPCRs are among the few molecules whose deletion not only alters synaptic transmission, but actually decrease synapse numbers, which suggests that these adhesion GPCRs are involved in synapse formation and maintenance (*Bolliger et al., 2011*; *Kakegawa et al., 2015*; *Anderson et al., 2017*; *Sando et al., 2019*). In particular, we recently observed that the conditional deletion of latrophilin-2 (Lphn2) from CA1-region pyramidal neurons caused a selective loss of synapses formed in the Stratum lacunosum-moleculare by inputs from the entorhinal cortex, whereas the conditional deletion of latrophilin-3 (Lphn3) caused a selective loss of synapses formed in the Stratum oriens and Stratum radiatum by Schaffer-collateral inputs from the CA3 region (*Anderson et al., 2017*; *Sando et al., 2019*). Importantly, rescue experiments involving overexpression of Lphn2 or Lphn3 revealed that Lphn2 and Lphn3 were non-redundant in the CA1-region (*Sando et al., 2019*). These two adhesion GPCRs could only rescue the phenotype of the deletion of the eponymous gene, but not of the deletion of the other gene, demonstrating an

*For correspondence:
tcs1@stanford.edu

unexpected functional diversification of highly homologous genes. These results not only demonstrated that latrophilins function as postsynaptic adhesion molecules, but also that they exhibit synapse specificity and are essential for the existence of their target synapses.

Here, we asked whether this conclusion reflects a general principle of latrophilin function. We identified robust expression of Lphn2 and Lphn3 in cerebellar Purkinje cells, and thus examined whether Lphn2 and Lphn3 perform individual non-redundant functions in these neurons as well. Unexpectedly, we found that unlike the function of Lphn2 and Lphn3 in hippocampal CA1-region neurons, Lphn2 and Lphn3 perform redundant functions in Purkinje cells. This function selectively targets parallel-fiber synapses, corroborating the synapse specificity observed in hippocampus. In addition, in our studies on the cerebellar expression of latrophilins we observed high expression levels of Lphn3 in Bergmann glia, a cerebellum-specific type of astrocyte. However, here unlike Purkinje cells we detected no functional changes in cerebellar physiology upon the Lphn3 deletion, suggesting that Lphn3 is not essential for Bergmann glia function.

## Results

### Differential expression of latrophilin genes in cerebellum

We first examined the expression of latrophilins in the cerebellum. The expression of latrophilins is developmentally regulated with peak expression observed at postnatal day P10, therefore we probed cerebellar sections at P10, P30 and P60 to cover latrophilin expression during synaptogenesis and in adult brain (*Sando et al., 2019*). Single molecule RNA fluorescent in situ hybridizations (smRNA-FISH) showed that the three latrophilin genes are differentially expressed in cerebellum, but in overlapping patterns (*Figure 1*). All latrophilins are expressed in cerebellar cortex; in addition, Lphn1, Lphn2 and to a lesser extent Lphn3 are expressed in deep cerebellar nuclei (*Figure 1A*). In the cerebellar cortex, Lphn1 is primarily expressed in granule cells, Lphn2 in Purkinje cells, and Lphn3 in Bergmann glia (that are also localized to the Purkinje cell layer) and in interneurons (*Figure 1B and C*). Quantification of Lphn RNA puncta in Purkinje cell bodies revealed that transcripts for all Lphn isoforms are found in Purkinje cells and that Lphn1 and Lphn2 puncta were more abundant than Lphn3 puncta (*Figure 1—figure supplement 1*).

In situ hybridizations identify key cell types with high-level gene expression, but can overlook lower levels of expression that might still be significant. To probe the cell-type specific expression of latrophilins by an independent method, we crossed Lphn2 and Lphn3 double cKO mice (*Sando et al., 2019*; *Anderson et al., 2017*) with five different Cre-driver mouse lines: Nestin-Cre mice that express Cre recombinase in the precursor cells which generate all cerebellar cell types, L7/*Pcp2*-Cre mice that express Cre recombinase only in Purkinje cells, PV-Cre mice that express Cre recombinase in parvalbumin-expressing cerebellar cells (Purkinje cells, inhibitory interneurons, and neurons of the deep cerebellar nuclei), Math1/*Atoh1*-Cre mice that express Cre recombinase in granule cells, and GLAST/*Slc1a3*-CreER mice that express tamoxifen-inducible Cre recombinase in Bergmann glia (*Figure 2A and B*). For the GLAST-CreER mice, Cre activity was induced by intraperitoneal injections of 4-hydroxytamoxifen on postnatal days P10, P11, and P12. All mice were analyzed on day P35. Lphn2 and Lphn3 protein levels were measured in the cerebellum by quantitative immunoblotting, using antibodies to HA and to GFP since the cKO mice express Lphn2 with an mVenus tag (*Anderson et al., 2017*) and Lphn3 with an HA-epitope (*Sando et al., 2019*).

Nestin-Cre nearly completely ablated Lphn2 and Lphn3 expression as expected (*Figure 2C*). L7-Cre had no significant effect on the levels of either Lphn2 or Lphn3, although there was a trend towards a decrease for Lphn2 (*Figure 2D*). This result seems to contradict the in situ data showing abundant expression of Lphn2 in Purkinje cells (*Figure 1B and C*). However, despite their remarkable size and prominence, the number of Purkinje cells is much lower than that of other cerebellar neurons (e.g., granule cells, interneurons, and deep cerebellar nuclei neurons), and the selective loss of Lphn2 and Lphn3 from Purkinje cells may have been occluded by the combined levels of these proteins in other neurons, in particular granule cells and deep cerebellar nuclei neurons. Consistent with this conclusion, both Lphn2 and Lphn3 were significantly decreased in PV-Cre mice that express Cre in deep cerebellar nuclei neurons and interneurons in addition to Purkinje cells, with Lphn2 expression reduced by nearly 80% (*Figure 2E*). Math-Cre, conversely, had no significant effect on the levels of Lphn2 and Lphn3, consistent with the fact that granule cells primarily express Lphn1 (*Figure 2F*).

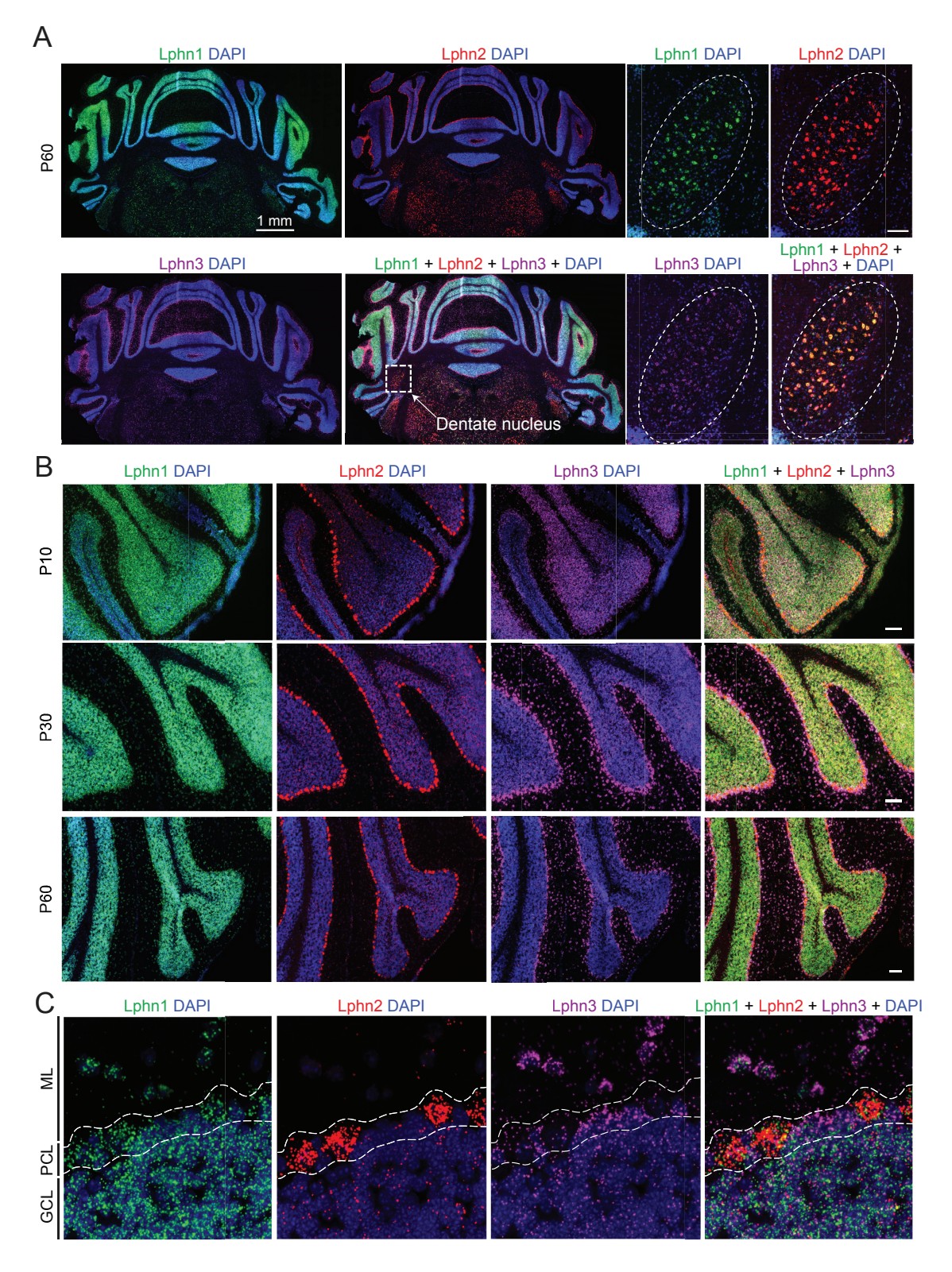

**Figure 1.** Differential expression of Lphn1, Lphn2, and Lphn3 mRNAs in cerebellum. (**A**) Overview of the expression of latrophilins in the cerebellum. smRNA-FISH was performed on mouse cerebellar sections at P60 with probes against Lphn1 (green), Lphn2 (red) and Lphn3 (purple) to visualize expression of latrophilin transcripts (left, overview of whole cerebellum; right, expanded images of the dentate nucleus, scale bar: 100 μm). (**B**) Time course of latrophilin expression in cerebellar cortex. smRNA-FISH for all three latrophilin isoforms was performed with mouse cerebellar sections at P10,

*Figure 1 continued on next page*

*Figure 1 continued*

P30 and P60 (scale bars: 100 μm). (C) Expanded view of the cerebellar cortex at P60 showing expression of latrophilin transcripts in the molecular layer (ML), Purkinje cell layer (PCL) and granule cell layer (GCL). Lphn1 and Lphn3 transcripts are detected in all three cerebellar layers shown, while Lphn2 transcripts are primarily identified in the Purkinje cell layer.

The online version of this article includes the following figure supplement(s) for figure 1:

**Figure supplement 1.** Lphn1 and Lphn2 transcripts are more abundant in Purkinje cell bodies than Lphn3 transcripts.

GLAST-Cre-mediated deletion of Lphn2 and Lphn3 in Bergmann glia, finally, did not detectably alter the levels of Lphn2, but dramatically decreased the levels of Lphn3, indicative of substantial Lphn3 expression in these astrocytes (*Figure 2G*).

## Deletions of either Lphn2 or Lphn3 alone from Purkinje cells does not significantly alter parallel-fiber synapses, whereas the double deletion of both Lphn2 and Lphn3 suppresses parallel-fiber synapses

Since latrophilins are postsynaptic adhesion molecules and Purkinje cells are the postsynaptic target of almost all synapses in the cerebellar cortex, we focused our analyses of the cerebellar role of latrophilins on Purkinje cells.

First, we examined individual Lphn2 and Lphn3 cKO mice, examining parallel-fiber synapses as the most abundant cerebellar synapse. We analyzed mice obtained from crosses of Lphn2 and Lphn3 cKO mice with L7-Cre mice and compared Cre-negative homozygous Lphn2 or Lphn3 cKO mice to Cre-positive littermate mice to assess the effects of the Lphn deletion. To estimate parallel-fiber synapses morphologically, we stained cerebellar sections for vGluT1. Because parallel-fiber synapses are too abundant in the cerebellar cortex to resolve individual vGluT1-positive puncta by confocal microscopy, we measured the overall vGluT1 staining intensity, which depends on both the density and the size of parallel-fiber synapses (*Zhang et al., 2015*). We observed no significant difference in vGluT1 staining intensity between sections from control mice and mice lacking either Lphn2 or Lphn3 in the Purkinje cells (*Figure 3A and E*). Next, we cut acute slices from the mice and used whole-cell patch-clamp recordings to measure the input resistance and capacitance of Purkinje cells, but again detected no difference between control and Lphn2- or Lphn3-deficient Purkinje cells (*Figure 3B and F*). Finally, we monitored parallel-fiber EPSCs induced by extracellular stimulation in the upper molecular layer, and determined both the paired-pulse ratio of EPSCs in response to two stimuli separated by 50 ms and the overall synaptic strength quantified by the input-output relation and the rise slope of EPSCs (*Figure 3C–D and G–H*). Here, we also observed no differences between Purkinje cells expressing or lacking Lphn2 or Lphn3, suggesting that each isoform alone is not essential for proper parallel-fiber synapse numbers or parallel-fiber synaptic transmission.

To address the potential for redundancy among Lphn2 and Lphn3, we generated double cKO mice in which both Lphn2 and Lphn3 are deleted in Purkinje cells by an L7-Cre allele. We stained cerebellar sections from littermate control and double cKO mice for vGluT1 to examine parallel-fiber synapses, for vGluT2 to assess climbing-fiber synapses and for vGAT to monitor inhibitory basket-cell and stellate-cell synapses (*Figure 4*). Strikingly, we found that the Lphn2/3 double-deficient sections displayed a ~ 30% decrease in vGluT1 staining intensity, suggesting a decrease in parallel-fiber synapses (*Figure 4A*). The density of vGluT2-positive puncta and of vGAT-positive puncta was unchanged (*Figure 4B and C*). The two types of excitatory inputs onto Purkinje cells can be distinguished anatomically as parallel fibers form synapses onto secondary and tertiary dendrites while climbing-fibers form synapses onto primary dendrites that are closer to the Purkinje cell soma. In support of a reduction of parallel-fiber but not climbing-fiber synapse number, we stained Lphn2/3 double-deficient sections for Homer1 and found that Homer1 puncta density was reduced on distal tertiary dendrites but not primary dendrites of Purkinje cells (*Figure 4—figure supplement 1*). In addition, we compared synaptic protein levels of cerebellar lysates from control and Lphn2/3 double-KO mice by immunoblot and found no difference in any of the proteins tested, including vGluT1 (*Figure 4D*). As discussed in our immunoblot analysis with the L7-Cre line in *Figure 2D,* a reduction in vGluT1 levels at Purkinje cell synapses in the cerebellar cortex may be obscured by abundant vGluT1 expression in other regions of the cerebellum. Overall, these results indicate that the double

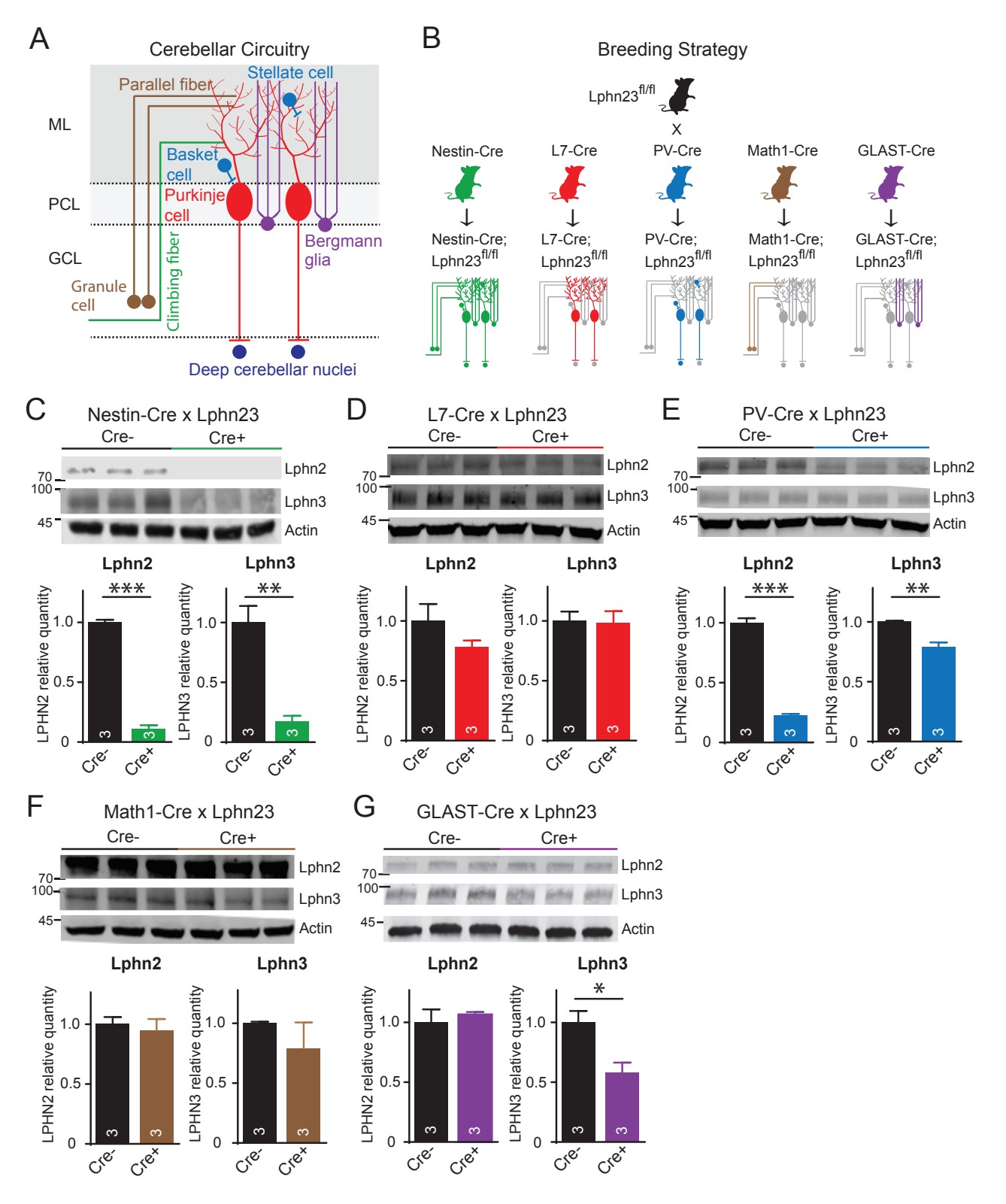

**Figure 2.** Cell type-specific deletion of Lphn2 and Lphn3 induce differential reduction in Lphn2 and Lphn3 protein levels. (**A**) Schematic of the cellular circuitry of the cerebellar cortex. Shown are parallel-fiber (brown) and climbing-fiber (green) excitatory inputs onto Purkinje cells, basket and stellate inhibitory interneurons (blue-green) in the molecular layer, Bergmann glia (purple) extending processes into the molecular layer and Purkinje cell outputs onto deep cerebellar nuclei (dark blue). (**B**) Breeding strategy for generation of cell type-specific knockouts of Lphn2 and Lphn3 using various

*Figure 2 continued on next page*

*Figure 2 continued*

transgenic Cre-driver mouse lines. The specific cell types targeted by the Cre lines are as follows: Nestin-Cre: all neurons and glia, L7-Cre: Purkinje cells, PV-Cre: molecular layer interneurons (basket and stellate cells), Purkinje cells and some deep cerebellar nuclei neurons, Math1-Cre: granule cells, GLAST-Cre: Bergmann glia. (C–G) Conditional knock-in mice with mVenus-tagged Lphn2 and HA-tagged Lphn3 were crossed with transgenic Cre-driver lines to examine the cell type-specific expression of Lphn2 and Lphn3 by immunoblotting in cerebellum of littermate mice with and without Cre recombinase expression (top, representative immunoblot images; bottom, summary graphs). (C) Expression of Lphn2 and Lphn3 in cerebellum is largely abolished in nestin-Cre positive mice, indicating that both are predominantly expressed in neurons and glia. (D) Lphn2 and Lphn3 expression in Purkinje cells is limited compared to their total cerebellar expression, although this may be confounded by the relative scarcity of Purkinje cells relative to other cell types in cerebellum. (E) Lphn2 (~80%) and Lphn3 (~20%) expression in PV-positive neurons make up a significant proportion of total Lphn cerebellar expression. (F) Lphn2 and Lphn3 expression in granule cells is limited relative to total cerebellar Lphn expression. (G) A significant proportion of Lphn3 (~50%) but not of Lphn2 in cerebellum is expressed in Bergmann glia, a cerebellar type of astrocyte. All numerical data are means ± SEM; numbers in bars represent number of mice tested. Statistical analyses were performed using Student's t-test (*$p<0.05$; **$p<0.01$; ***$p<0.001$).

Lphn2/3 deletion may specifically alter parallel-fiber synapses, suggesting a redundant function of Lphn2 and Lphn3 in these synapses.

We further tested the functional impact of the double Lphn2/3 deletion using whole-cell patch-clamp recordings from control and Lphn2/3 double KO Purkinje cells. Lphn2/3-deficient and control Purkinje cells exhibited the same capacitance and input resistance values, suggesting that the double deletion did not grossly impair the development of Purkinje cells (*Figure 5A*). Input-output measurements of parallel-fiber synaptic responses, however, uncovered an approximately 50% decrease in parallel-fiber synaptic strength, confirming the immunohistochemistry findings that suggest that parallel-fiber synapses are impaired by the double Lphn2/3 deletion (*Figure 5B*). This decrease is likely not due to a decrease in release probability, as demonstrated by a lack of change in paired-pulse ratios (*Figure 5C*). Analyses of climbing-fiber synaptic responses showed that the amplitude, paired-pulse ratio, and step number of climbing-fiber EPSCs (reflecting the number of climbing fibers innervating a Purkinje cell) were normal in Lphn2/3 double-deficient Purkinje cells (*Figure 5D*).

Recordings of spontaneous mEPSCs performed in the presence of tetrodotoxin revealed no differences in the average mEPSC frequency between Lphn2/3 double-deficient and control Purkinje cells (*Figure 5E–F*). mEPSC rise times can be used to distinguish between events likely derived from parallel-fiber inputs that form synapses onto distal dendrites of Purkinje cells and events likely derived from climbing-fiber inputs that form synapse onto proximal Purkinje cell dendrites. The majority of mEPSCs with 10–90% rise times of less than 0.6 ms are derived from climbing fibers that are insensitive to Group III mGluR agonists, while the majority of mEPSCs with 10–90% rise times greater than 2 ms are derived from parallel-fiber synapses that are severely depressed in response to Group III mGluR agonists (*Ichikawa et al., 2016*). The frequency of mEPSCs with rise times less than 0.6 ms was unchanged between Lphn2/3 double cKO and control cells, whereas the frequency of mEPSCs with rise times greater than 2 ms was reduced by ~25%, a difference that was almost statistically significant ($p=0.07$). The frequency of mEPSCs with rise times between 0.6 ms and 2 ms, which reflects a mixed population of parallel-fiber- and climbing-fiber-derived events, was only slightly reduced in Lphn2/3 deficient Purkinje cells. A significant difference in the cumulative frequency distribution of mEPSC amplitudes was observed despite minimal difference in average mEPSC amplitude after the Lphn2/3 deletion (*Figure 5G*). The average 10–90% rise times of mEPSCs was significantly reduced in Lphn2/3 deficient Purkinje cells compared to control, however, the average rise times for both the <0.6 ms and >2 ms subgroups of mEPSCs were unchanged, suggesting that the reduction in average mEPSC rise time was the result of the altered proportion of parallel-fiber to climbing-fiber events after Lphn2/3 double KO rather than a change in intrinsic mEPSC properties of either synaptic input (*Figure 5H*). Recordings of spontaneous miniature inhibitory postsynaptic currents (mIPSCs) revealed no changes in mIPSC amplitude or frequency between control and Lphn2/3-deficient Purkinje cells (*Figure 5I*). Overall, these data indicate that the double deletion of both Lphn2 and Lphn3, as opposed to the single deletion of each latrophilin isoform individually, causes a selective and robust impairment of parallel-fiber synapses.

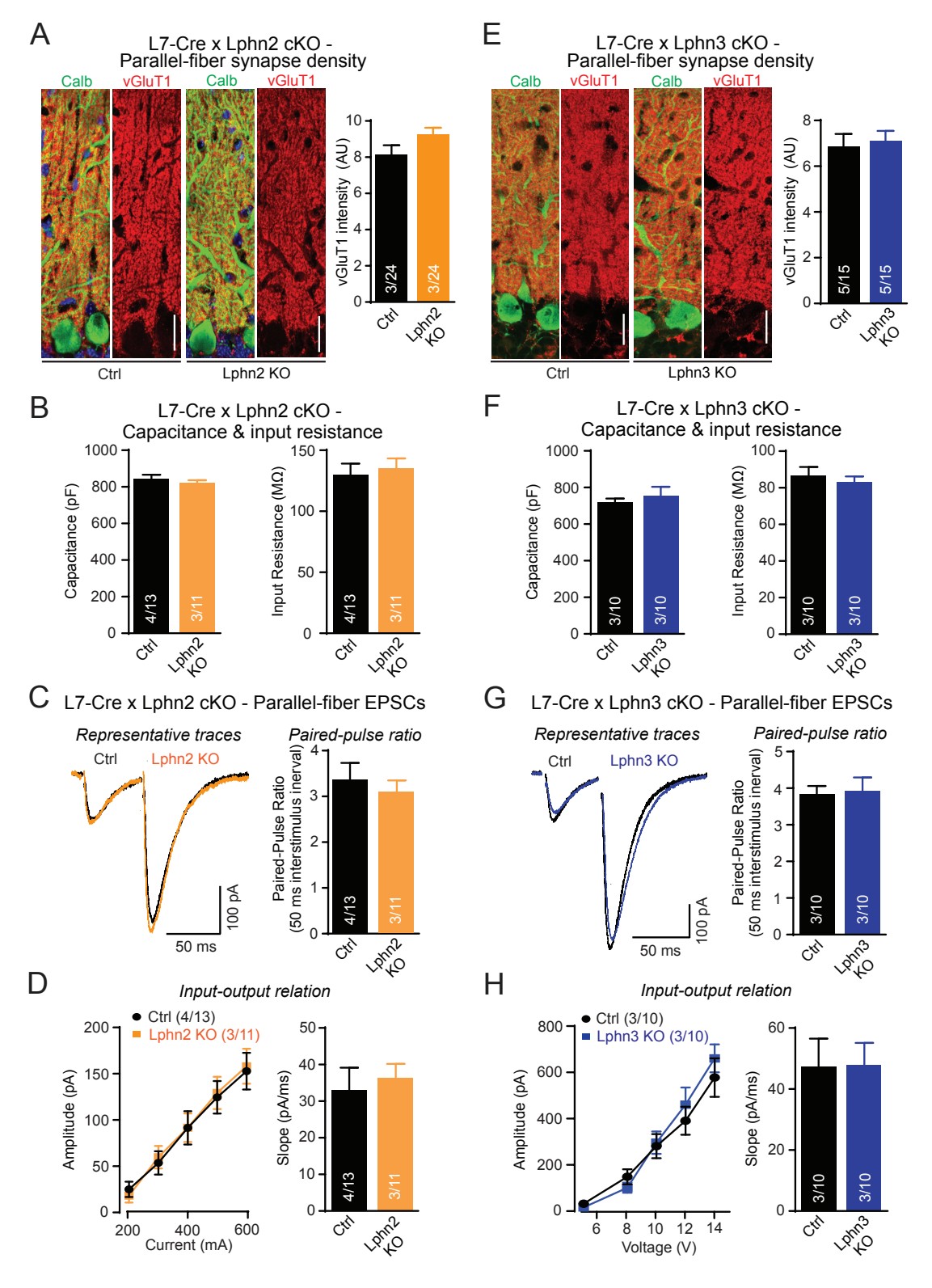

**Figure 3.** Individual deletions of Lphn2 or Lphn3 in Purkinje cells do not detectably affect parallel-fiber synapse density or function. (**A**) Lphn2 deletion from Purkinje cells does not affect parallel-fiber synapse staining as shown by unchanged vGluT1 staining intensity. vGluT1 specifically labels parallel-fiber synapses and vGluT1 staining intensity serves as a proxy for parallel-fiber synapse density (left, representative images with vGluT1 in red and calbindin in green; right, summary graph). (**B**) Purkinje cell capacitance (left) and input resistance (right) are unchanged by Lphn2 KO. (**C–D**) Lphn2

*Figure 3 continued on next page*

*Figure 3 continued*

deletion does not affect evoked parallel-fiber synaptic transmission measured by recording parallel-fiber excitatory postsynaptic currents (parallel-fiber EPSCs) from Purkinje cells evoked by electrical stimulation in the upper molecular layer. The paired-pulse ratio (PPR, (**C**), input-output relation and EPSC slope (**D**) are unchanged by Lphn2 deletion (C left, representative traces with 50 ms inter-stimulus interval; C right, summary graph of PPR; D, summary graphs of input-output relation and EPSC slope). (**E**) Lphn3 KO does not affect parallel-fiber synapse density (as in A). (**F**) Purkinje cell capacitance (left) and input resistance (right) are unchanged by Lphn3 KO. (**G–H**) Same as (**C-D**) but for the Lphn3 KO, demonstrating that the Lphn3 deletion from Purkinje cells also does not affect parallel-fiber synaptic transmission. All scale bars are 20 µm. Numerical data are means ± SEM; numbers in bars represent independent experiments/number of cells tested.

## Despite high levels of expression, Lphn3 is not essential in Bergmann glia for cerebellar synapses

Given the selective high-level expression of Lphn3 in Bergmann glia in the cerebellum we observed by in situ hybridization analyses (*Figures 1* and *2*) that is consistent with single-cell RNAseq data (*Saunders et al., 2018*; *Zeisel et al., 2018*; *Zhang et al., 2014*; *Koirala and Corfas, 2010*), we asked whether Lphn3 might have a synaptic organizing function in these cells. This question was prompted by our unpublished finding that the expression of neuroligin-3, another postsynaptic adhesion molecule, in Bergmann glia appears to be essential for the function of parallel-fiber synapses (Zhang, B. and Südhof, T.C., submitted). Thus, we used Lphn3 cKO mice crossed with GLAST-CreER mice to delete Lphn3 from astrocytic Bergmann glia in the cerebellum, employing daily 4-OHT injections from age P10-P12 to induce Cre recombination (*Figure 6A*). This manipulation caused a major decrease in Lphn3 protein levels as determined by immunoblot in total cerebellum (*Figure 2G*). Consistent with this observation, we observed a similar decrease in Lphn3 staining intensity (visualized by the HA-epitope knock-in) in the cerebellar cortex after deletion of Lphn3 in Bergmann glia (*Figure 6B*). These results indicate that the majority of Lphn3 in cerebellar cortex is expressed by Bergmann glia, and that despite the functional importance of Lphn3 in Purkinje cells as described above, the Lphn3 levels in Purkinje cells are much lower than those in Bergmann glia.

We first examined markers for excitatory parallel-fiber and climbing-fiber synapses as well as for inhibitory synapses to determine if Lphn3 deletion from Bergmann glia impairs synapses of the cerebellar cortex (*Figure 6C–E*). As above, we quantified parallel-fiber synapses by overall vGluT1 staining intensity because the density of parallel-fiber synapses is too high to enable resolution of individual puncta by confocal microscopy. The resulting vGluT1 staining intensity reflects a combination of parallel-fiber synapse density and size (*Zhang et al., 2015*). The deletion of Lphn3 from Bergmann glia had no effect on this parameter (*Figure 6C*). We then quantified the density, staining intensity, and size of vGluT2-positive puncta that represent climbing-fiber synapses. We observed no change in climbing-fiber density, but a significant decrease in their staining intensity and size, hinting at a possible impairment (*Figure 6D*). Moreover, we investigated inhibitory synapses both in the upper molecular and the Purkinje cell layer that are presumably derived from stellate cells and basket cells, respectively, but again observed no significant changes (*Figure 6E*).

We next asked whether the deletion of Lphn3 from Bergmann glia might change the presence and composition of Bergmann glia processes in the cerebellar cortex. We used immunocytochemistry to stain the cerebellar cortex of littermate-control homozygous Lphn3 cKO mice containing or lacking a GLAST-Cre allele for three Bergmann glia-specific markers: the glutamate transporters GLAST and GLT-1, and the glutamate receptor GluA1 (*Figure 6F*; *Rothstein et al., 1994*; *Bergles et al., 1997*; *Saab et al., 2012*). For every glial marker, the staining pattern intensity were indistinguishable between sections from control mice and mice in which Lphn3 had been deleted from Bergmann glia postnatally, suggesting that Bergmann glia were unaffected by Lphn3 deletion (*Figure 6F*).

These experiments suggest that the Lphn3 deletion from Bergmann glia does not cause a major change in the structure and synaptic composition of the cerebellar cortex. To independently confirm this conclusion, we analyzed the cerebellum from control mice and mice with a deletion of Lphn3 from Bergmann glia by quantitative immunoblotting for a series of synaptic marker proteins (*Figure 6G*). We observed no changes in any synaptic protein, in particular not in vGluT1 and vGluT2 that are specific for parallel-fiber and climbing-fiber synapses, respectively, except for a decrease in the levels of vGAT that is specific for inhibitory synapses, and a trend towards a decrease for the scaffolding proteins CASK and gephyrin. These results confirm that the deletion of Lphn3 from

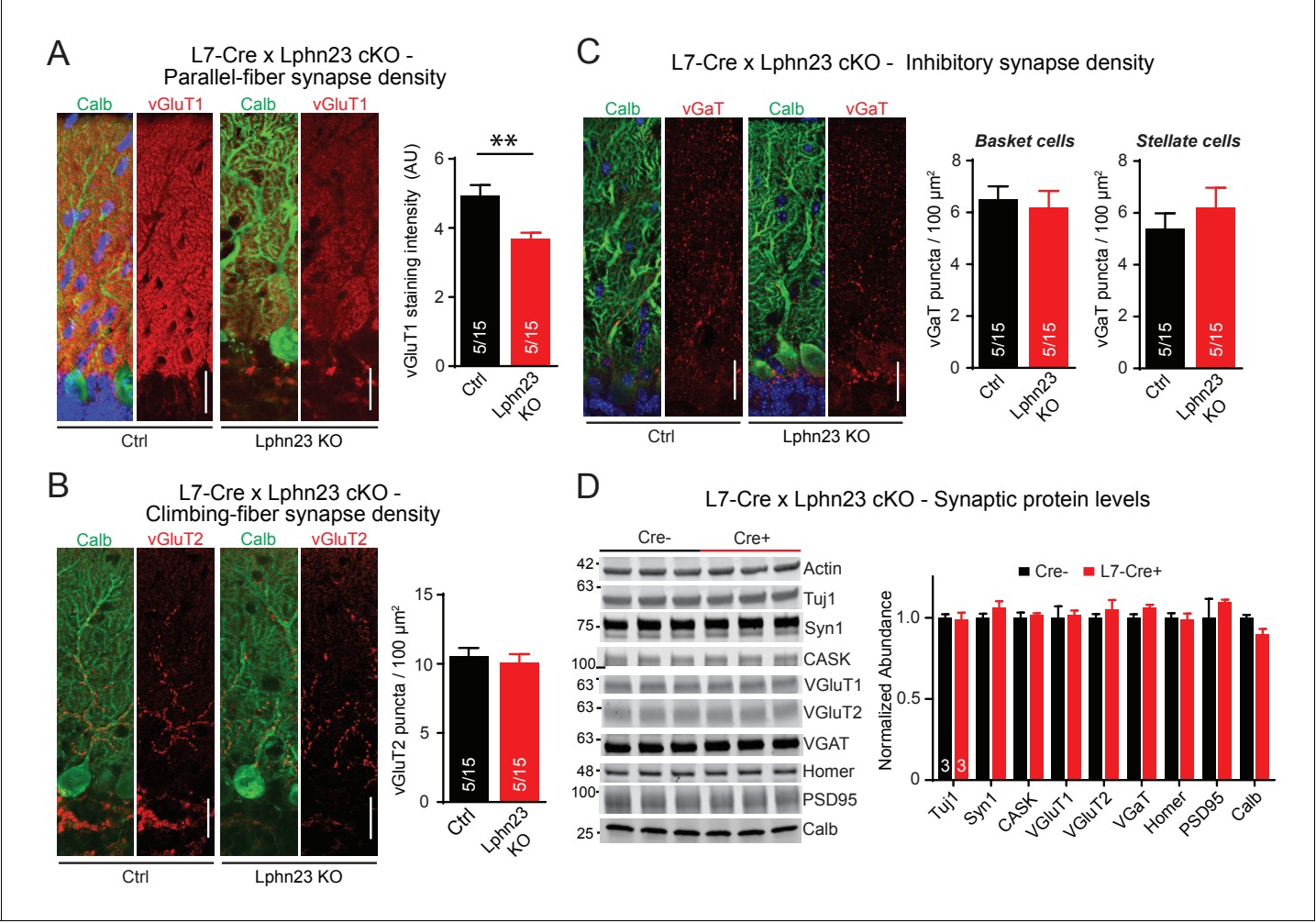

**Figure 4.** Double deletion of Lphn2 and Lphn3 in Purkinje cells decreases parallel-fiber synapse staining but has no effect on climbing-fiber synapse density, inhibitory synapse density in the cerebellar cortex or synaptic protein composition of the cerebellum. (A) Deletion of both Lphn2 and Lphn3 from Purkinje cells reduces parallel-fiber synapse staining as shown by reduced vGluT1 staining intensity that specifically labels parallel-fiber synapses and has been previously shown to be a proxy for parallel-fiber synapse density (left, representative images of vGluT1 in red and calbindin in green; right, summary graph). (B) The Lphn2/3 double deletion from Purkinje cells does not alter climbing-fiber synapse density as visualized by vGluT2 puncta staining. Climbing-fiber synapse density was quantified by counting vGluT2 puncta (red) co-stained with calbindin (green) as a function of Purkinje cell dendrite area (left, representative images; right, summary graph). (C) The Lphn2/3 double deletion from Purkinje cells does not alter inhibitory basket-cell or stellate-cell synapse density as shown by vGaT puncta staining. Basket-cell synapse density was quantified by measuring vGaT puncta density (red) on Purkinje cell bodies identified by calbindin staining (green), while stellate-cell synapse density was quantified by measuring vGaT puncta density in the upper half of the molecular layer (left, representative images; right, summary graph). (D) The Lphn2/3 double deletion from Purkinje cells does not alter synaptic protein levels from whole cerebellum (top, representative immunoblots; bottom, summary graph of protein abundance normalized to actin loading controls, n = 3 each). All scale bars are 20 μm. Data are represented as means ± SEM; numbers in bars in A, B, C represent independent experiments/number of sections imaged. **p<0.01 (two-tailed t test).

The online version of this article includes the following figure supplement(s) for figure 4:

**Figure supplement 1.** Double deletion of Lphn2 and Lphn3 from Purkinje cells decreases Homer1 puncta on distal tertiary but not primary Purkinje cell dendrites.

Bergmann glia, the major cell type in cerebellum in which Lphn3 is expressed, does not cause a major change in the structure and composition of the cerebellar cortex.

In a final set of experiments we explored the possibility that the Lphn3 deletion from Bergmann glia may impair cerebellar synaptic functions even though the overall synaptic architecture of the cerebellum appears to be largely unaffected by the Lphn3 deletion. Using whole-cell patch-clamp recordings from Purkinje cells in acute cerebellar slices, we confirmed by measurements of the

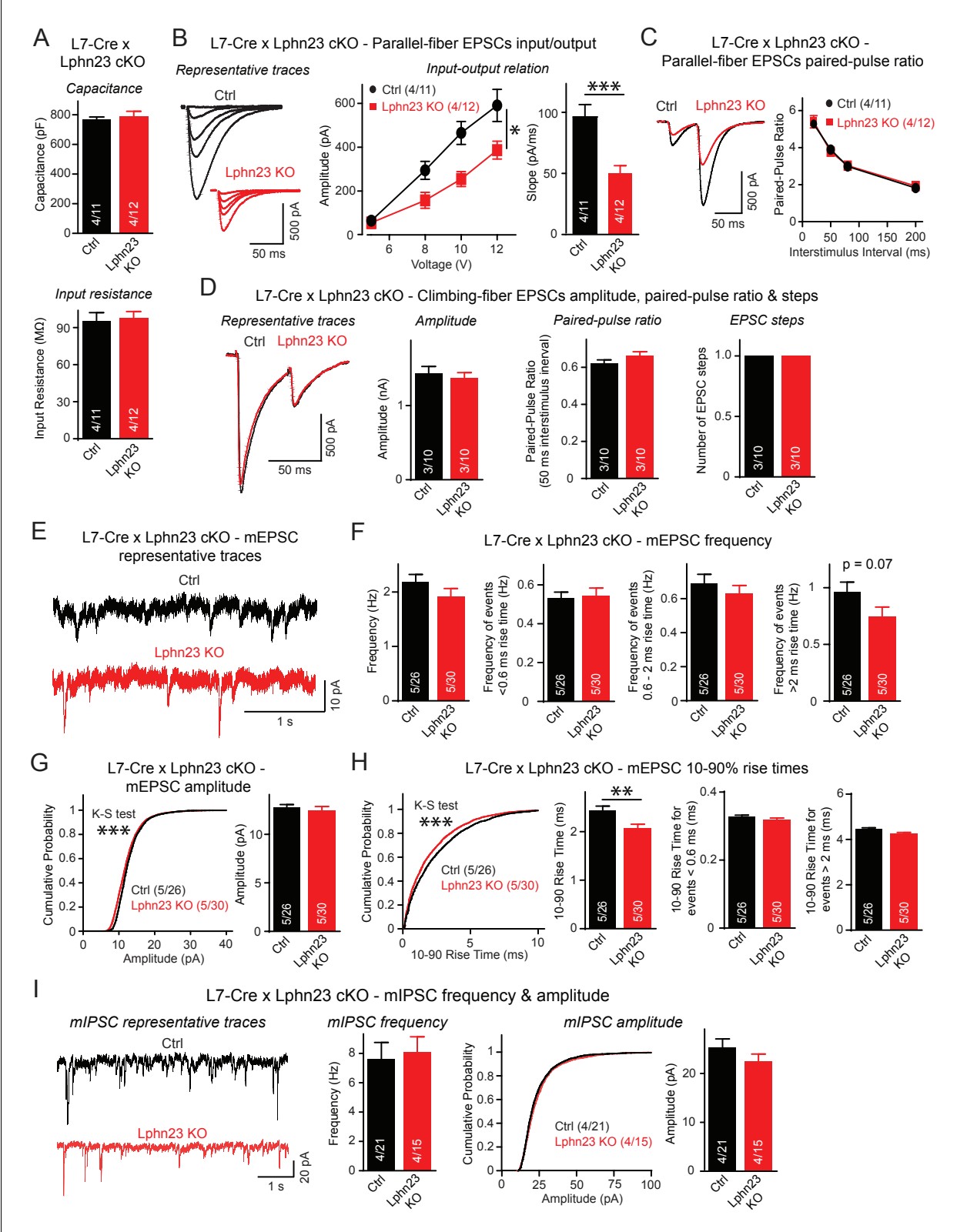

**Figure 5.** Double deletion of Lphn2 and Lphn3 from Purkinje cells suppresses parallel-fiber synaptic transmission without altering climbing-fiber synaptic transmission. (**A**) The Lphn2/3 double deletion from Purkinje cells does not affect their capacitance or input resistance. (**B**) The Lphn2/3 double deletion from Purkinje cells impairs parallel-fiber synaptic transmission as shown by diminished input-output relation and reduced slope of parallel-fiber excitatory postsynaptic currents (EPSCs) recorded from Purkinje cells (left, representative traces showing parallel-fiber EPSCs at increasing stimulus

*Figure 5 continued on next page*

*Figure 5 continued*

intensities; right, summary graph of input-output relation and parallel-fiber EPSC slope). (C) The Lphn2/3 double deletion from Purkinje cells has no effect of the paired-pulse ratio (PPR) of parallel-fiber EPSCs (left, representative traces of parallel-fiber-EPSCs of identical stimulus intensity with a 50 ms inter-stimulus interval; right, summary graph of PPRs with varying inter-stimulus intervals). (D) The Lphn2/3 double deletion from Purkinje cells does not affect climbing-fiber synaptic transmission or pruning with no differences observed in climbing-fiber EPSC amplitude, PPR or the number of steps in the evoked climbing-fiber response (left, representative traces of climbing-fiber EPSCs with a 50 ms inter-stimulus interval; right, summary graphs of climbing-fiber EPSC amplitude, PPR and number of steps). (E–F) The Lphn2/3 double deletion from Purkinje cells does not alter the frequency of spontaneous miniature excitatory postsynaptic currents (mEPSCs) but reduces the frequency of mEPSCs with rise times greater than 2 ms (E, representative traces of mEPSCs; F, summary graphs of mEPSC frequency of all events and events separated by 10–90% rise times). A majority of mEPSCs with 10–90% rise times under 0.6 ms are derived from climbing-fibers which synapse onto proximal Purkinje cell dendrites, while a majority mEPSCs with 10–90% rise times greater than 2 ms arise from parallel-fibers which synapse onto distal Purkinje cell dendrites. A trend nearing significance was observed for lower frequency of mEPSCs with 10–90% rise times greater than 2 ms in Lphn2/3 double KO cells compared to control (p=0.07). (G) The Lphn2/3 double deletion from Purkinje cells has minimal effects on mEPSC amplitude (left, cumulative frequency relation of mEPSC amplitudes; right, summary graph). A significant difference in the cumulative frequency relation of mEPSC amplitudes was observed between Lphn2/3 double KO and control by the Kolmogorov–Smirnov test but no significant difference in average mEPSC amplitude was identified by a two-tailed t test. (H) The Lphn2/3 double deletion from Purkinje cells reduces the 10–90% rise time of mEPSCs (left, cumulative frequency relation of rise times; right, summary graphs of all rise times, rise times less than 0.6 ms and rise times greater than 2 ms). Both the average and cumulative frequency relation of 10–90% rise times were significantly different between Lphn2/3 double KO and control, but no difference in average 10–90% rise times was detected in the subsets of mEPSCs with rise times under 0.6 ms or greater than 2 ms. (I) The Lphn2/3 double deletion from Purkinje cells does not affect amplitude or frequency of spontaneous miniature inhibitory post-synaptic currents (mIPSCs) recorded from Purkinje cells (left, representative traces of mIPSCs; middle, summary graph of mIPSC frequency; right, cumulative frequency plot and summary graph of mIPSC amplitudes). Data are means ± SEM; numbers in bars represent independent experiments/number of cells tested. *p<0.05, **p<0.01, ***p<0.001 (two-tailed t test and two-way ANOVA).

capacitance and input resistance that the Lphn3 deletion in Bergmann glia has no major effect on the basic properties of Purkinje cells (*Figure 7A*). Evoked climbing-fiber synaptic responses were unchanged in either amplitude, paired-pulse ratio or number of 'steps' (reflecting the number of climbing fibers innervating a Purkinje cell) after Lphn3 deletion from Bergmann glia (*Figure 7B*). Measurements of spontaneous miniature EPSCs and IPSCs (performed in the presence of tetrodotoxin) showed that the frequency and amplitude of mEPSCs and of mIPSCs were also unchanged (*Figure 7C–D*). Thus, despite high levels of Lphn3 expression in Bergmann glia, Lphn3 is not essential in this type of astrocyte for any of the major synaptic components of the cerebellar cortex.

## Discussion

Latrophilins are adhesion GPCRs that were recently revealed to perform a key role in hippocampal synapse formation. Specifically, we showed that in pyramidal CA1-region neurons, postsynaptic Lphn2 is selectively essential for excitatory synapses formed onto the distal dendritic domains of these neurons in the Stratum lacunosum-moleculare (*Anderson et al., 2017*), whereas Lphn3 is selectively essential for excitatory synapses formed onto the proximal dendritic domains in the Stratum oriens and Stratum radiatum (*Sando et al., 2019*). In hippocampal CA1-region neurons, individual deletions of Lphn2 and Lphn3 produce robust but distinct phenotypes, and Lphn2 overexpression does not rescue the Lphn3 deletion phenotype, demonstrating that the two latrophilin isoforms are not functionally redundant in these neurons. It was thus surprising for us to find in the current experiments that the individual deletions of Lphn2 and Lphn3 in cerebellar Purkinje cells produced no significant phenotypes, but that the double deletion of both Lphn2 and Lphn3 caused a large selective impairment of parallel-fiber synapses. Based on these results, we conclude that although Lphn2 and Lphn3 are functionally distinct in CA1-region pyramidal neurons, they are functionally redundant in cerebellar Purkinje cells.

The effect of the Lphn2/3 double deletion on Purkinje cells is selective for parallel-fiber synapses as evidenced by immunocytochemistry for specific synaptic markers (*Figure 4*) and by electrophysiological recordings of evoked and spontaneous synaptic transmission (*Figure 5*). The most parsimonious explanation for the impairment of parallel-fiber synapses in Lphn2/3-deficient Purkinje cells is that there is a decrease in synapse numbers because the vGluT1 staining intensity in the cerebellar cortex was decreased and the strength of parallel-fiber synapses was suppressed without a change in paired-pulse ratio. In addition, the frequency of long (>2 ms) rise time mEPSC events largely derived from parallel-fiber synapses was decreased while the frequency of short (<0.6 ms) rise time

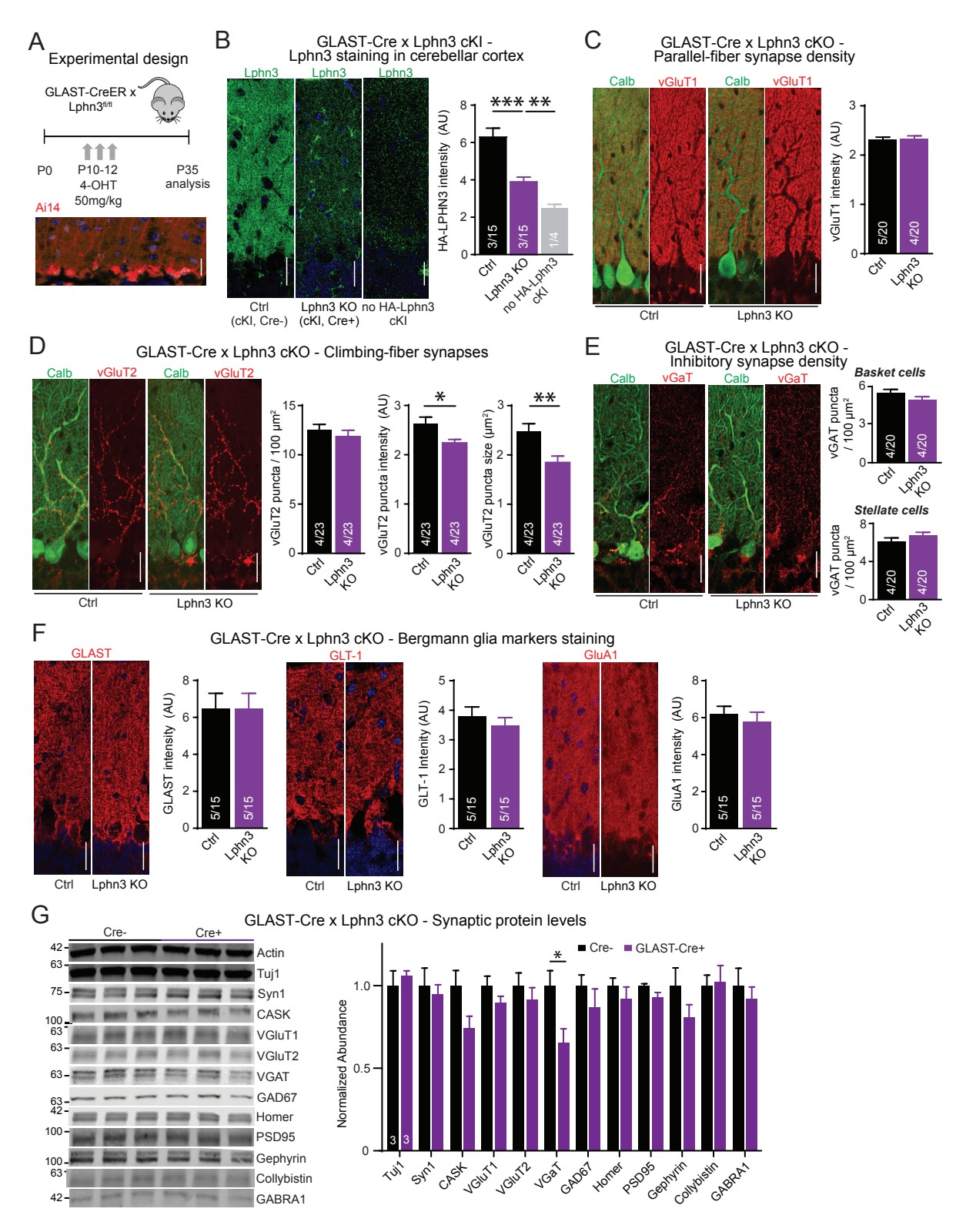

**Figure 6.** Deletion of Lphn3 in Bergmann glia has minimal effects on parallel-fiber, climbing-fiber or inhibitory synapse staining in the cerebellar cortex, Bergmann glia staining intensity, or synaptic protein composition of the cerebellum. (**A**) Schematic of experimental procedure. Lphn3 cKO mice were crossed to the GLAST-CreER line to allow for tamoxifen-inducible Cre recombinase expression in glia. Mice were injected with 50 mg/kg of 4-hydroxytamoxifen (4-OHT) once daily from P10-12 and analysis was performed at P35. Bottom image shows a cerebellar section from a GLAST-CreER

*Figure 6 continued on next page*

Figure 6 continued

mouse with an Ai14 (Cre-dependent tdTomato) allele treated with the same induction protocol, demonstrating widespread Cre expression in Bergmann glia. (B) Lphn3 deletion from Bergmann glia reduces the intensity of HA-Lphn3 staining in the molecular layer of cerebellar sections (left, representative images; right, summary graph). Because of significant background staining in wild-type sections, the analysis shown compares staining for HA of HA-Lphn3 knock-in sections lacking Cre with those of the HA-Lphn3 knock-in with Bergmann glia deletion and of a control lacking the HA-Lphn3 knock-in allele. (C–E) Lphn3 deletion from Bergmann glia does not change parallel-fiber (C), climbing-fiber (D) or molecular-layer interneuron (E) synapse density in cerebellar cortex as visualized by vGluT1, vGluT2 and vGaT staining, respectively. Sections were double labeled for vGluT1, vGluT2, or vGAT (red) and calbindin (green, to label Purkinje cells). In (E) a significant decrease in vGluT2 puncta intensity and size was observed. (F) Lphn3 deletion from Bergmann glia does not alter the levels and distribution of three Bergmann glial marker proteins important for glial function, GLAST, GLT-1 and GluA1 (red). (G) Deletion of Lphn3 from cerebellar Bergmann glia has little effect on the synaptic protein composition of the cerebellum (left, representative immunoblots of a panel of synaptic proteins in cerebellar lysates; right, summary graph of normalized synaptic protein abundance, n = 3 each). Expression levels were normalized first to actin then to Cre negative controls. For all panels (C to F), representative images are shown on the left and summary graphs on the right. All scale bars are 20 µm. Data are means ± SEM; numbers in bars represent independent experiments/ number of sections imaged. *p<0.05, **p<0.01, ***p<0.001 (one-way ANOVA and two-tailed t test).

events largely derived from climbing-fiber synapses was unchanged. In this respect, the Lphn2/3 double deletion phenotype resembles that of the Lphn2 and Lphn3 single deletions in CA1-region neurons of the hippocampus in which we also observed a loss of synapses without a change in release probability that was restricted to specific subset of excitatory synapses (*Anderson et al., 2017*; *Sando et al., 2019*). The L7 promoter used for Purkinje cell-specific Cre recombination initiates Cre expression at P9-10, which is during the period of parallel-fiber synaptogenesis (*Barski et al., 2000*; *Mishina et al., 2012*). While the current study does not distinguish between a role for Lphns in synapse formation versus synapse maintenance, excitatory synaptic transmission onto CA1 neurons of the hippocampus is similarly impaired by Lphn3 deletion at P0 and P21, suggesting that Lphns may have a broad role in excitatory synapse maintenance.

Many synaptic adhesion molecules, including neurexins and neuroligins, are expressed in glia (*Saunders et al., 2018*; *Zeisel et al., 2018*). Our quantitative immunoblotting and smRNA-FISH data show that Lphn3 is expressed in Bergmann glia at extraordinarily high levels, as evidenced by the fact that the Lphn3 deletion from Bergmann glia caused a ~ 50% decrease in cerebellar Lphn3 protein content (*Figure 2G*). Nevertheless, deletion of latrophilin-3 from Bergmann glia had no major effect on Bergmann glia markers or synaptic parameters in the cerebellum (*Figures 6–7*). Although we observed a small decrease in climbing-fiber synapse puncta size and intensity by immunohistochemistry, climbing-fiber synaptic transmission was unchanged, suggesting that Lphn3 expression in Bergmann glia is not essential for the fundamental properties of these cerebellar synapses. This lack of a phenotype is unexpected especially if one compares the effect of the Lphn3 deletion when induced in combination with the Lphn2 deletion in Purkinje cells in the analyses of the Lphn2/3 double-deficient condition to that of the Lphn3 deletion in Bergmann glia. In Purkinje cells, Lphn3 is barely detectable by in situ hybridization and the deletion of Lphn3 from Purkinje cells has no significant effect on the overall cerebellar Lphn3 content; nevertheless, its deletion from Purkinje cells causes a massive impairment in parallel-fiber synapses. In Bergmann glia, in contrast, Lphn3 levels are so high that they contribute ~50% to the total cerebellar Lphn3 content even with widespread Lphn3 expression elsewhere in cerebellum, but here deletion of Lphn3 is without major consequences. Our findings do not preclude a role for Lphn3 in Bergmann glia earlier in development, as the deletion of Lphn3 studied here was initiated at P10, similar to the start of Cre expression in the L7-Cre mouse line. This example illustrates again a conclusion that has previously been proposed many times, namely that the expression levels of a protein are not necessarily indicative of its functional importance in a given cell type.

Finally, one may ask why the phenotype of the Lphn2 and Lphn3 deletions, although very significant and similar in magnitude to that of the cerebellin-1 deletion (*Ito-Ishida et al., 2008*), is not even stronger at parallel-fiber synapses. It seems likely that there may be further redundancy between Lphn1, the latrophilin isoform that was not examined here, with Lphn2 and Lphn3. Lphn1 is highly expressed in many cerebellar cell types (*Figure 1*), but no cKO mice are currently available. Given the redundancy between Lphn2 and Lphn3 observed here, presumably Lphn1 may also be

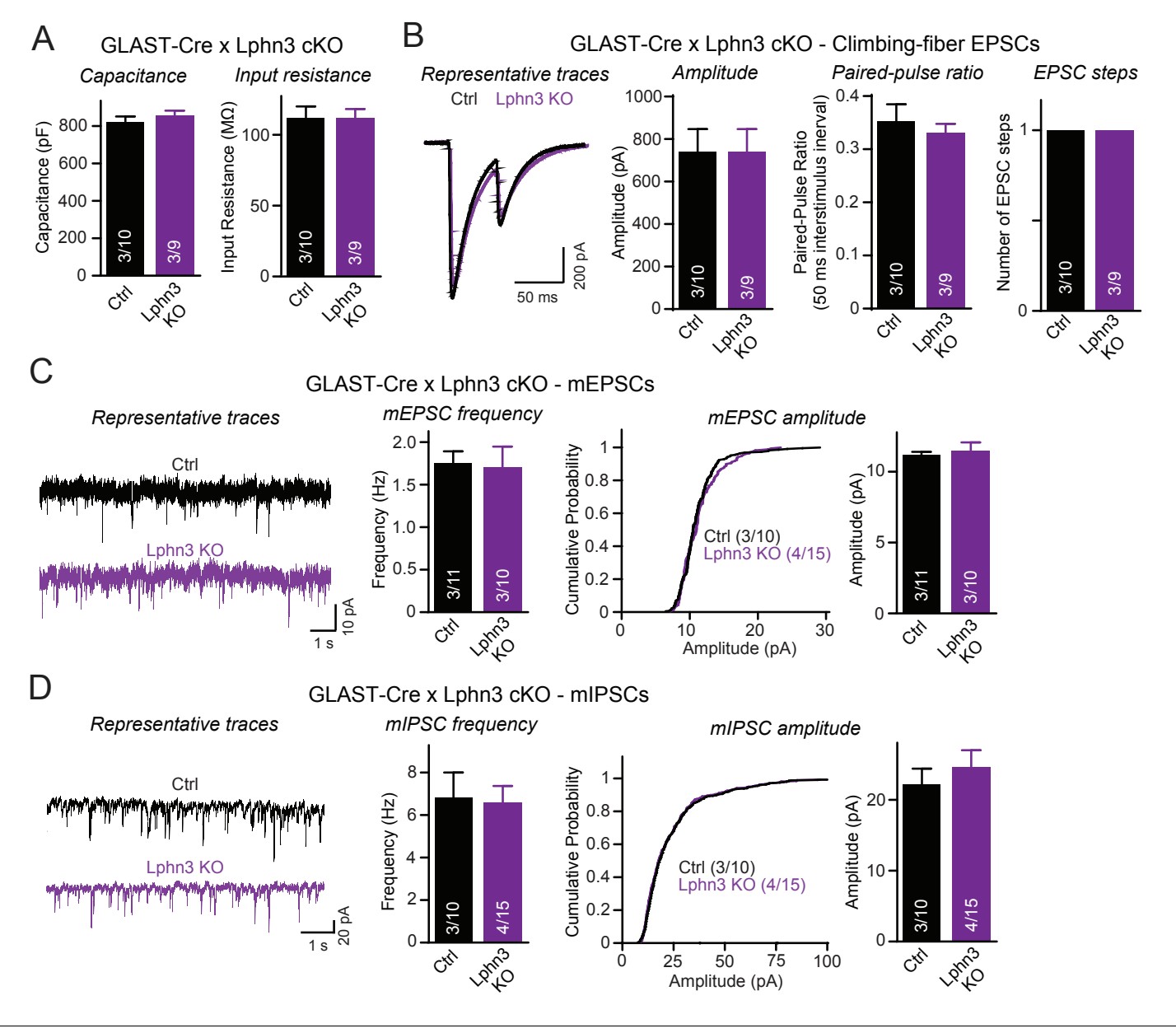

**Figure 7.** Deletion of Lphn3 in Bergmann glia does not alter climbing-fiber synaptic transmission or spontaneous Purkinje cell miniature currents. (**A**) Deletion of Lphn3 from Bergmann glia does not affect the Purkinje cell capacitance or input resistance. (**B**) Climbing-fiber synaptic transmission is unchanged by the deletion of Lphn3 from Bergmann glia, with no difference in climbing-fiber EPSC amplitude, paired-pulse ratio or number of steps in the EPSC response observed (left, representative traces of climbing-fiber EPSCs with a 50 ms inter-stimulus interval; right, summary graphs of climbing-fiber EPSC amplitude, PPR and number of steps). (**C**) Lphn3 deletion from Bergmann glia has no effect on the amplitude and frequency of spontaneous mEPSCs recorded from Purkinje cells (left, representative traces of mEPSCs; middle, summary graph of mEPSC frequencies; right, cumulative frequency plot and summary graph of mEPSC amplitudes). (**D**) Lphn3 deletion from Bergmann glia has no effect on the amplitude and frequency of spontaneous mIPSCs recorded from Purkinje cells (left, representative traces of mIPSCs; middle, summary graph of mIPSC frequencies; right, cumulative frequency plot and summary graph of mIPSC amplitudes). Data are means ± SEM; numbers in bars represent independent experiments/number of cells tested.

redundant with Lphn2 and Lphn3, and a triple Lphn1/2/3 deletion could potentially have a much bigger impact on synaptic function than a double deletion of Lphn2 and Lphn3.

In summary, our results reveal a selective redundant function of Lphn2 and Lphn3 as postsynaptic adhesion molecules in parallel-fiber synapses of the cerebellum. These results extend the previous conclusion obtained in hippocampal CA1 region neurons that Lphn2 and Lphn3 are essential postsynaptic adhesion molecules, but demonstrate that although Lphn2 and Lphn3 perform distinct functions in these hippocampal neurons, they perform redundant functions in Purkinje cells.

# Materials and methods

## Key resources table

| Reagent type (species) or resource | Designation | Source or reference | Identifiers | Additional information |
|---|---|---|---|---|
| Strain, strain background (*Mus musculus*) | Lphn2 cKO mice | The Jackson Laboratory | RRID:IMSR_JAX:023401 | |
| Strain, strain background (*Mus musculus*) | Lphn3 cKO mice | The Jackson Laboratory | RRID:IMSR_JAX:026684 | |
| Strain, strain background (*Mus musculus*) | L7-Cre mice | The Jackson Laboratory | RRID:IMSR_JAX:004146 | |
| Strain, strain background (*Mus musculus*) | Nestin-Cre mice | The Jackson Laboratory | RRID:IMSR_JAX:003771 | |
| Strain, strain background (*Mus musculus*) | Math1-Cre mice | The Jackson Laboratory | RRID:IMSR_JAX:011104 | |
| Strain, strain background (*Mus musculus*) | GLAST-CreERT2 mice | The Jackson Laboratory | RRID:IMSR_JAX:012586 | |
| Strain, strain background (*Mus musculus*) | PV-Cre mice | The Jackson Laboratory | RRID:IMSR_JAX:017320 | |
| Strain, strain background (*Mus musculus*) | Ai14 mice | The Jackson Laboratory | RRID:IMSR_JAX:007914 | |
| Sequence-based reagent | Lphn1 RNA FISH probe | Advanced Cell Diagnostics | Cat. #: 319331 | |
| Sequence-based reagent | Lphn2 RNA FISH probe | Advanced Cell Diagnostics | Cat. #: 319341-C2 | |
| Sequence-based reagent | Lphn3 RNA FISH probe | Advanced Cell Diagnostics | Cat. #: 317481-C3 | |
| Antibody | IRDye 680LT donkey polyclonal anti rabbit | Licor | 926–68023, RRID:AB_10706167 | 1:10000 |
| Antibody | IRDye 800CW donkey polyclonal anti mouse | Licor | 926–32212, RRID:AB_10715072 | 1:10000 |
| Antibody | IRDye 680LT donkey polyclonal anti guinea pig | Licor | 926–68030, RRID:AB_10706310 | 1:10000 |
| Antibody | Anti-HA (mouse monoclonal) | Covance | MMS-101P, RRID:AB_2565007 | 1:5000 WB, 1:1000 IHC |
| Antibody | Anti-GFP (rabbit polyclonal) | Invitrogen | A11122, RRID:AB_221569 | 1:5000 |
| Antibody | Anti-ACTB (mouse monoclonal) | Sigma | AC-15, RRID:AB_476692 | 1:10000 |
| Antibody | Anti-Tuj1 (mouse monoclonal) | Covance | MMS-435P, RRID:AB_2313773 | 1:5000 |

*Continued on next page*

*Continued*

| Reagent type (species) or resource | Designation | Source or reference | Identifiers | Additional information |
|---|---|---|---|---|
| Antibody | Anti-synapsin-1 (mouse monoclonal) | Sysy | 106011, RRID:AB_2619772 | 1:5000 |
| Antibody | Anti-CASK (mouse monoclonal) | Neuromab | K56A/50, RRID:AB_2750669 | 1:5000 |
| Antibody | Anti-vGluT1 (guinea pig polyclonal) | Millipore | AB5905, RRID:AB_2301751 | 1:5000 WB, 1:1000 IHC |
| Antibody | Anti-vGluT2 (guinea pig polyclonal) | Millipore | AB2251, RRID:AB_1587626 | 1:5000 WB, 1:1000 IHC |
| Antibody | Anti-vGaT (rabbit polyclonal) | Millipore | AB5062, RRID:AB_2301998 | 1:5000 WB, 1:1000 IHC |
| Antibody | Anti-GAD67 (mouse monoclonal) | Millipore | MAB5406, RRID:AB_2278725 | 1:5000 |
| Antibody | Anti-Homer1 (rabbit polyclonal) | Sysy | 160003, RRID:AB_887730 | 1:5000 |
| Antibody | Anti-Homer1 (rabbit polyclonal) | Millipore | ABN37, RRID:AB_11203298 | 1:1000 |
| Antibody | Anti-PSD95 (mouse monoclonal) | Sysy | 124011, RRID:AB_10804286 | 1:5000 |
| Antibody | Anti-gephyrin (mouse monoclonal) | Neuromab | L106/83, RRID:AB_2617120 | 1:5000 |
| Antibody | Anti-collybistin (mouse monoclonal) | Neuromab | L120/12, RRID:AB_2728741 | 1:5000 |
| Antibody | Anti-GABARA1 (mouse monoclonal) | Neuromab | N95/35, RRID:AB_2700770 | 1:5000 |
| Antibody | Anti-calbindin (mouse monoclonal) | Sigma | C9848, RRID:AB_476894 | 1:5000 WB, 1:2000 IHC |
| Antibody | Anti-GLAST (rabbit polyclonal) | Abcam | Ab416, RRID:AB_304334 | 1:500 |
| Antibody | Anti-GLT-1 (rabbit polyclonal) | Abcam | Ab41621, RRID:AB_941782 | 1:500 |
| Antibody | Anti-GluA1 (rabbit polyclonal) | Sysy | 182003, RRID:AB_2113441 | 1:500 |
| Antibody | Anti-MAP2 (chicken polyclonal) | Encor Biotech | CPCA-MAP2, RRID:AB_2138173 | 1:1000 |
| Antibody | Alexa fluor 488, goat polyclonal anti mouse IgG | Invitrogen | RRID:AB_2535711 | 1:1000 |
| Antibody | Alexa fluor 488, goat polyclonal anti chicken IgG | Invitrogen | RRID:AB_2534096 | 1:1000 |
| Antibody | Alexa fluor 546, goat polyclonal anti guinea pig IgG | Invitrogen | RRID:AB_2534118 | 1:1000 |
| Antibody | Alexa fluor 546, goat polyclonal anti rabbit IgG | Invitrogen | RRID:AB_2534093 | 1:1000 |
| Antibody | Alexa fluor 633, goat polyclonal anti guinea pig IgG | Invitrogen | RRID:AB_2535757 | 1:1000 |
| Chemical compound, drug | Picrotoxin | Tocris | 1128 | |
| Chemical compound, drug | CNQX | Tocris | 1045 | |

*Continued on next page*

*Continued*

| Reagent type (species) or resource | Designation | Source or reference | Identifiers | Additional information |
|---|---|---|---|---|
| Chemical compound, drug | AP5 | Tocris | 0106 | |
| Chemical compound, drug | Tetrodotoxin citrate | Tocris | 0640 | |
| Software, algorithm | Clampfit | Molecular Devices | RRID:SCR_011323 | |
| Software, algorithm | pClamp | Molecular Devices | RRID:SCR_011323 | |
| Software, algorithm | Prism | Graphpad Software Inc | RRID:SCR_002798 | |
| Software, algorithm | Image Studio Lite | Licor | RRID:SCR_014211 | |
| Software, algorithm | NIS-Elements | Nikon | RRID:SCR_002776 | |
| Software, algorithm | Odyssey CLx | Licor | RRID:SCR_014579 | |

## Mouse generation and husbandry

Lphn2 and Lphn3 conditional knockout (cKO) and knock-in lines were generated as described (*Sando et al., 2019*; *Anderson et al., 2017*). Other mouse lines used in this paper were L7/*Pcp2*-Cre (*Barski et al., 2000*; JAX #004146), PV-Cre (JAX #017320), Math1/*Atoh1*-Cre (*Matei et al., 2005*; JAX #011104), GLAST/*Slc1a3*-CreERT2 (JAX #012586), Ai14 (JAX #007914) and Nestin-Cre (*Tronche et al., 1999*, JAX #003771). All mice used were on a CD57BL/6 background. All procedures conformed to the National Institutes of Health Guidelines for the Care and Use of Laboratory Animals and were approved by the Stanford University Administrative Panel on Laboratory Animal Care. All analyses of Lphn KOs were performed with male and female littermate-controlled Cre-positive and Cre-negative mice that were homozygous for single or double Lphn cKOs.

## Single-molecule RNA fluorescent in situ hybridization

Wild type CD-1 mice were euthanized with isofluorane at P10, P30 and P60 followed by transcardial perfusion with ice cold PBS. Brains were quickly dissected and embedded in Optimal Cutting Temperature (OCT) solution on dry ice. 16 µm thick sections were cut using a Leica CM3050-S cryostat, mounted directly on to Superfrost Plus slides and stored at −80°C until use. Single-molecule FISH for Lphn1 (Cat# 319331), Lphn2 (Cat# 319341-C2) and Lphn3 (Cat# 317481-C3) mRNA was performed using the multiplex RNAscope platform (Advanced Cell Diagnostics) according to manufacturer instructions for fresh-frozen sections.

## Immunoblotting

Immunoblotting was performed as described previously (*Zhang et al., 2015*). Littermate-controlled mice at P35 were anesthetized and cerebella were dissected out, then homogenized in lysis buffer containing (in mM): 20 Tris-HCl, pH 7.5, 100 NaCl, 4 KCl, 2 MgCl2, 2 CaCl2, 1% Triton X-100, 1x protease inhibitor cocktail (Roche) at 4°C. The samples were then centrifuged for 20 mins at 20,000 g to remove insoluble materials at 4°C. 20 µg of protein lysate was run on 4–20% SDS-polyacrylamide gels then transferred onto nitrocellulose membranes on a semi-dry transfer apparatus (BioRad Trans-Blot Turbo). Membranes were blocked in 5% milk powder in 0.05% TBS-Tween for 1 hr at room temperature (RT). Primary antibodies were diluted in blocking buffer and incubated overnight at 4°C. Membranes were then washed four times for 10 min with 0.05% TBS-Tween then incubated with fluorescence-labeled secondary antibodies (donkey anti-rabbit IR Dye 680CW, 1:10,000; donkey anti-mouse IR Dye 800CW, 1:10,000; donkey anti-guinea pig IR Dye 680CW, 1:10,000; all from LI-COR Biosciences). Signals were visualized with an Odyssey Infrared Imager and Odyssey software (LI-COR Biosciences). Total intensity values were calculated with Odyssey software and quantification was performed by normalizing protein abundance

to actin first then to Cre-negative control samples. Antibodies used: anti-HA, 1:5000, Covance, MMS-101P; anti-GFP, 1:5000, rabbit, Invitrogen, A11122; anti-ACTB, 1:10000, mouse, Sigma, AC-15; anti-Tuj1, 1:5000, mouse, Covance, MMS-435P; anti-synapsin 1, 1:5000, mouse, Sysy, 106011; anti-synapsin 1, 1:5000, rabbit, Yenzym, clone YZ6079; anti-CASK, 1:5000, mouse, Neuromab, K56A/50; anti-vGluT1, 1:5000, guinea pig, Millipore, AB5905; anti-vGluT2, 1:5000, guinea pig, Millipore, AB2251; anti-vGaT, 1:5000, rabbit, Millipore, AB5062; anti-GAD67, 1:5000, mouse, Millipore, MAB5406; anti-Homer1, 1:5000, rabbit, Sysy, 160003; anti-Homer1, 1:5000, rabbit, Yenzym, YZ6081; anti-PSD95, 1:5000, mouse, Sysy, 124011; anti-gephyrin, 1:5000, mouse, Neuromab, L106/83; anti-collybistin, 1:5000, mouse, Neuromab, L120/12; anti-GABARA1, 1:5000, mouse, Neuromab, N95/35; anti-calbindin, 1:5000, mouse, Sigma, C9848.

## Immunohistochemistry

Immunohistochemistry was performed as previously described (*Zhang et al., 2015*). Mice (P21 for L7-Cre, P35 for GLAST-CreER KOs) were anesthetized with isoflurane, perfused with phosphate buffered saline (PBS) to clear blood then with 4% paraformaldehyde (PFA) in 0.1M PBS to fix tissues. Cerebella were dissected out and post-fixed in 4% PFA overnight at 4°C. Cerebella were then cryoprotected in 30% sucrose in 0.1 M PBS for a minimum of 48 hr at 4°C. Sagittal cerebellar sections (40 μm) were collected at −20°C with a cryostat (Leica CM1050). Sections were washed with PBS and incubated in blocking buffer (0.2% Triton X-100% and 5% goat serum in PBS) for 1 hr at RT under gentle agitation, followed by incubation for 24 hr at 4°C with primary antibodies diluted in blocking buffer (antibodies used: anti-vGluT1, 1:1000, guinea pig, Millipore, AB5905; anti-vGluT2, 1:1000, guinea pig, Millipore, AB2251; anti-vGaT, 1:1000, rabbit, Millipore, AB5062; anti-calbindin, 1:2000, mouse, Sigma, C9848; anti-GLAST, 1:500, rabbit, Abcam, ab416s; anti-GLT-1, 1:500, rabbit, Abcam, ab41621; anti-GluA1, 1:500, rabbit, Sysy, 182003; anti-HA, 1:1000, mouse, Covance, MMS-101P; anti-Homer1, 1:1000, rabbit, Millipore, ABN37; anti-MAP2, 1:1000, chicken, Encor Bioscience, CPCA-MAP2). Sections were washed 4 times for 10 min each in PBS then incubated in secondary antibodies diluted in blocking buffer for 1 hr at RT (1:1000, Alexa 488 or 546, Invitrogen). Sections were washed as described previously then mounted onto positively-charged microscope slides and covered in DAPI-containing mounting media (Vectashield, Vector Labs). Serial confocal z-stack images (1.5 μm intervals for 40 μm at 1024 × 1024 resolution) were acquired using a Nikon confocal microscope (A1Rsi) with a 60x oil objective (PlanApo, NA1.4). All staining procedures and acquisition parameters were kept constant within each experiment. Images of cerebellum molecular layer were taken from the midpoint of lobule IV/V. Image backgrounds were normalized, and immunoreactive elements were analyzed with Nikon analysis software (object size range 0.25–4.0 μm$^2$).

## Electrophysiology

Electrophysiological recordings from cerebellum were performed as previously described (*Zhang et al., 2015*; *Caillard et al., 2000*; *Llano et al., 1991*). Mice at age P21-25 or P35-40 were anesthetized with isofurane and decapitated. Cerebella were rapidly extracted and transferred into ice-cold low-Ca2+ aCSF containing (in mM): 125 NaCl, 2.5 KCl, 3 MgCl2, 0.1 CaCl2, 25 glucose, 1.25 NaH2PO4, 0.4 ascorbic acid, three myo-inositol, 2 Na-pyruvate, and 25 NaHCO3; pH was adjusted to 7.4 by continuous gassing with carbogen. Sagittal slices (250 μm) were sectioned from the vermis on a vibratome (Leica VT 1200S) and transferred to oxygenated aCSF at RT containing (in mM): 1 MgCl2, 2 CaCl2 instead of 3 MgCl2, 0.1 CaCl2 as described earlier. Slices were incubated for >1 hr before recording. Purkinje cells were visually identified using infrared DIC imaging with a 40 × water immersion objective. Whole-cell recordings from Purkinje cells in cerebellar lobules IV/V (voltage-clamped at −70 mV) were performed at RT with borosilicate glass pipettes (2–3 MΩ) pulled with a vertical micropipette puller (PC-10, Narishige). Internal pipette solutions contained (in mM): for all EPSCs, 140 Cs-gluconate, 5 CsCl, 2 MgCl2, 0.5 EGTA, 2 Na-ATP, 0.5 Na-GTP (pH 7.3, adjusted with CsOH); for IPSCs, 145 CsCl, 2 MgCl2, 0.5 EGTA, 2 Na-ATP, 0.5 NaGTP (pH 7.3, adjusted with CsOH). For CF-EPSC and PF-EPSC recordings, the aCSF bath solution contained 50 μM picrotoxin and 10 μM APV; miniature EPSC recordings also included 1 μM TTX. For mIPSC recordings, the aCSF bath solution contained 10 μM CNQX, 10 μM APV and 1 μM TTX. Focal square pulse stimuli (duration 50 μs, amplitude 5–15 V or 200-600mA) were applied with a bipolar stimulation electrode (FHC, Bowdoinham, ME) connected to a Model 2100 Isolated Pulse Stimulator (A-M

Systems, Inc) synchronized with the Clampfit 10 data acquisition software (Molecular Devices), with currents recorded by a Multiclamp 700B amplifier (Molecular Devices). Climbing fibers were stimulated in the granule cell layer around Purkinje cells (~100 µm) and identified by their characteristic all-or-none response and paired-pulse depression at a 50 ms inter-stimulus interval (ISI). The number of steps of climbing-fiber EPSCs was defined as the maximal number of climbing-fiber EPSC responses with different amplitudes recorded with different stimulus strengths at different stimulation sites in the granule cell layer. Parallel fibers were stimulated in the distal molecular layer (~200 µm from the target Purkinje cell) and identified by paired-pulse facilitation at 50 ms ISI. Only Purkinje cells with a series resistance <12 MΩ (not compensated; monitored before and after experiments by applying 10 ms, −5 mV voltage pulses) were used for analyses. Amplitude and PPR of evoked EPSCs were averaged from 5 to 10 traces. mEPSCs and mIPSCs were thresholded at 5 pA and hand-picked and analyzed using Clampfit 10.7 (Molecular Devices). Whole-cell currents were recorded at 10 kHz and filtered with a Bessel filter at 4 kHz.

## Data analyses and statistics

All data are shown as means ± SEM, with statistical significance (*$p<0.05$, **$p<0.01$, ***$p<0.001$) determined by Student's t-test, one-way analysis of variance (ANOVA), two-way ANOVA or Kolmogorov-Smirnov (K-S) tests. Non-significant results ($p>0.05$) are not specifically identified.

## Acknowledgements

We thank Dr. R Sando for advice on mouse genetics. This study was supported by a grant from the NIH. RSZ was supported by the Lucille P Markey Basic Biomedical Research Fellowship from Stanford University. K L-A is supported by EMBO long-term fellowship (ALTF 803–2017).

## Additional information

### Funding

| Funder | Grant reference number | Author |
|---|---|---|
| National Institute of Mental Health | R37-MH052804 | Thomas C Südhof |
| Stanford University | Lucille P. Markey Basic Biomedical Research Fellowship | Roger Shen Zhang |
| EMBO | ALTF 803-2017 | Kif Liakath-Ali |

The funders had no role in study design, data collection and interpretation, or the decision to submit the work for publication.

### Author contributions

Roger Shen Zhang, Conceptualization, Data curation, Formal analysis, Validation, Investigation, Visualization, Methodology, Writing - original draft, Writing - review and editing; Kif Liakath-Ali, Investigation, Visualization; Thomas C Südhof, Conceptualization, Supervision, Funding acquisition, Writing - original draft, Project administration, Writing - review and editing

### Author ORCIDs

Roger Shen Zhang (iD) https://orcid.org/0000-0003-2022-4004
Kif Liakath-Ali (iD) https://orcid.org/0000-0001-9047-7424
Thomas C Südhof (iD) https://orcid.org/0000-0003-3361-9275

### Ethics

Animal experimentation: All animal procedures conformed to the National Institute of Health Guidelines for the Care and Use of Laboratory Animals and were approved by the Stanford University Administrative Panel on Laboratory Animal Care (protocol #21589).

Decision letter and Author response
Decision letter https://doi.org/10.7554/eLife.54443.sa1
Author response https://doi.org/10.7554/eLife.54443.sa2

## Additional files

### Supplementary files
• Transparent reporting form

### Data availability
All data generated during this study are included in the manuscript.

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
