## [Decision Letter]

**Acceptance summary:**

Latrophillins are a fascinating set of molecules, first identified as the target of a toxin (α-latrotoxin) and more recently demonstrated to have endogenous ligands. They are expressed throughout the nervous system and there has been considerable interest in defining their function over the past 20 years. It appears that one of the major stumbling blocks may be the partial redundancy of function when more than one of these proteins is expressed in a postsynaptic neuron. This is highlighted here and overcome with the use of conditional, double knockouts, a definitive and clear approach.

**Decision letter after peer review:**

Thank you for submitting your article "Latrophilin-2 and Latrophilin-3 Are Redundantly Essential for Parallel-Fiber Synapse Function in Cerebellum" for consideration by *eLife*. Your article has been reviewed by two peer reviewers, including Graeme W Davis as the Reviewing Editor and Reviewer #1, and the evaluation has been overseen Ronald Calabrese as the Senior Editor. The following individual involved in review of your submission has agreed to reveal their identity: Wei Xu (Reviewer #2).

The reviewers have discussed the reviews with one another and the Reviewing Editor has drafted this decision to help you prepare a revised submission.

Summary:

This is an interesting and important paper from the Sudhof laboratory examining the role of GPCR adhesions molecules, the Latrophilins, in the cerebellum. Latrophillins are a fascinating set of molecules, first identified as the target of a toxin (α-latrotoxin) and more recently demonstrated to have endogenous ligands. They are expressed throughout the nervous system and there has been considerable interest in defining their function over the past 20 years. It appears that one of the major stumbling blocks may be the partial redundancy of function when more than one of these proteins is expressed in a postsynaptic neuron. This is highlighted here and overcome with the use of conditional, double knockouts, a definitive and clear approach.

The work follows on recent work demonstrating remarkable synapse-specific effects of these molecules within the hippocampal network. In the present work, latrophilin-1 and latrophilin-2 are demonstrated to both be required for the parallel fibers contacting postsynaptic PCs, without affecting the climbing fiber synapses. This was most clearly demonstrated electrophysiologically, a result that was anatomically confirmed as a decrease in vGlut staining. The phenotype is most consistent with a change in synapse number, as there were no changes in release dynamics, and given the decrease in vGlut staining.

The experiments are clear and nicely presented. In general, the reviewers agree that the work is appropriate for publication in *eLife*, appealing to the broad readership of *eLife*.

Essential revisions:

1) There was discussion regarding anatomical data supporting a conclusion of altered excitatory synapse number. It is acknowledged that the specific geometry of the excitatory synapses being studied is challenging for approaches that have been used previously, including retrograde tracing. However, considerable weight is given to vGlut staining. Could the authors expand upon their existing argument with additional semi-quantitative anatomical probes (other antibodies to pre- or postsynaptic diagnostic proteins)? Alternatively, would it be possible apply basic biochemical assays for synaptic protein abundances? In essence, the electrophysiological phenotype is clear. Can the proposed anatomical correlate be strengthened for excitatory synapses?

2) Could the authors comment directly on the developmental progression of CRE expression in the cerebellum for the CRE-drivers used in this study? And, by extension, could this information be used to infer the function of Lphn2/3 in the different stages of synapse development (formation, maturation versus stability)? Comments in the Discussion would be sufficient.

---

## [Author Response]

Essential revisions:1) There was discussion regarding anatomical data supporting a conclusion of altered excitatory synapse number. It is acknowledged that the specific geometry of the excitatory synapses being studied is challenging for approaches that have been used previously, including retrograde tracing. However, considerable weight is given to vGlut staining. Could the authors expand upon their existing argument with additional semi-quantitative anatomical probes (other antibodies to pre- or postsynaptic diagnostic proteins)? Alternatively, would it be possible apply basic biochemical assays for synaptic protein abundances? In essence, the electrophysiological phenotype is clear. Can the proposed anatomical correlate be strengthened for excitatory synapses?

We have performed an additional experiment at the request of the reviewers to support the finding that reduced vGluT1 staining and decreased mEPSC frequency is suggestive of reduced parallel-fiber synapse number in Lphn23 double cKO mice with the L7-Cre line (Figure 4—figure supplement 1). In the experiment, we stained cerebellar sections from Lphn23 double cKO and littermate control mice for Homer1, a postsynaptic density protein, in addition to vGluT1 to label parallel-fiber presynaptic terminals and MAP2 to label neuronal dendrites. We quantified the density of Homer1 puncta on both primary Purkinje cell dendrites that primarily receive climbing-fiber synaptic inputs and distal tertiary dendrites that primarily receive parallel-fiber inputs. We found that the density of Homer1 puncta on primary Purkinje cell dendrites was unchanged by Lphn23 double-deletion but that the density of Homer1 puncta on distal tertiary Purkinje cell dendrites was significantly reduced after Lphn23 double-deletion from Purkinje cells (Figure 4—figure supplement 1B-C). The ~20% reduction in Homer1 puncta density at distal dendrites is consistent with the magnitude of the reduction in mEPSC frequency with rise times > 2 ms that are primarily derived from parallel-fiber synapses. Homer1 puncta size and fluorescent intensity were unchanged by the Lphn23 double-deletion, but vGluT1 staining intensity at Homer1 puncta on distal dendrites was reduced in the Lphn23 double-deficient sections.

We believe this experiment provides additional anatomical support for the finding of reduced synapse number specifically at parallel-fiber inputs with Lphn23 double-deletion from Purkinje cells.

We have amended the Results to add the following statement:

“The two types of excitatory inputs onto Purkinje cells can be distinguished anatomically as parallel fibers form synapses onto secondary and tertiary dendrites while climbing-fibers form synapses onto primary dendrites that are closer to the Purkinje cell soma. In support of a reduction of parallel-fiber but not climbing-fiber synapse number, we stained Lphn2/3 double-deficient sections for Homer1 and found that Homer1 puncta density was reduced on distal tertiary dendrites but not primary dendrites of Purkinje cells (Figure 4—figure supplement 1).”

2) Could the authors comment directly on the developmental progression of CRE expression in the cerebellum for the CRE-drivers used in this study? And, by extension, could this information be used to infer the function of Lphn2/3 in the different stages of synapse development (formation, maturation versus stability)? Comments in the Discussion would be sufficient.

We have amended the Discussion to add the following statements at the suggestion of the reviewers.

“The L7 promoter used for Purkinje cell-specific Cre recombination initiates Cre expression at P9-10, which is during the period of parallel-fiber synaptogenesis (Barski et al., 2000; Mishina et al., 2012). While the current study does not distinguish between a role for Lphns in synapse formation versus synapse maintenance, excitatory synaptic transmission onto CA1 neurons of the hippocampus is similarly impaired by the Lphn3 deletion at P0 and P21, suggesting that Lphns may have a broad role in excitatory synapse maintenance.”

“Our findings do not preclude a role for Lphn3 in Bergmann glia earlier in development, as the deletion of Lphn3 studied here was initiated at P10, similar to the start of Cre expression in the L7-Cre mouse line.”